# Short-term health effects of an urban regeneration programme in deprived neighbourhoods of Barcelona

**Xavier Bartoll-Roca**[1,2]*, **María José López**[1,2,3,4], **Katherine Pérez**[1,2,3], **Lucía Artazcoz**[1,2,3,4], **Carme Borrell**[1,2,3,4]

**1** Agència de Salut Pública de Barcelona (ASPB), Barcelona, Catalunya, Spain, **2** Institut d'Investigació Biomèdica Sant Pau (IIB SANT PAU), Barcelona, Spain, **3** CIBER Epidemiología y Salud Pública (CIBERESP), Spain, **4** Universitat Pompeu Fabra, Barcelona, Spain

* xbartoll@aspb.cat

**Data Availability Statement:** The Barcelona Health Survey (BHS) forms part of the statistical actions of interest to the Generalitat de Catalunya and is included in the Annual Statistical Action

## Abstract

Urban regeneration programmes are interventions meant to enhance the wellbeing of residents in deprived areas, although empirical evidence reports mixed results. We evaluated the health impact of a participatory and neighbourhood-wide urban regeneration programme, Pla de Barris 2016–2020, in Barcelona. A pre-post with a comparison group study design. Using data from a cross-sectional survey performed in 2016 and 2021. The health outcomes analysed were mental health, alcohol and psychotropic drug use, perceived health status, physical activity and obesity. Depending on the investment, two intervention groups were defined: moderate- and high-intensity intervention groups. The analysis combined difference-in-difference estimation with an inverse weighting derived from a propensity score to reduce potential biases. The impact of the intervention in percentages and its confidence interval were estimated with a linear probability model with clustered adjusted errors. The intervention had a positive impact on health outcomes in women in the high-intensity intervention group: a reduction of 15.5% in the relative frequency of those experiencing poor mental health, and of 21.7% in the relative frequency of those with poor self-perceived health; and an increase of 13.7% in the relative frequency of those doing physical activity. No positive impact was observed for men, but an increase of 10.3% in the relative frequency of those using psychotropic drugs in the high-intensity intervention group. This study shows positive short-term effects of the urban regeneration programme Pla de Barris 2016–2020 on health outcomes in women in the high-intensity intervention group. These results can guide future interventions in other areas.

## Introduction

Urban regeneration programmes are complex interventions aimed at enhancing wellbeing in disadvantaged communities [1]. Looking for long-term advances, policy-makers are changing the focus of these programmes to improving the built and social environments [2]. This way,

Programme (PAAE) under the registration number: 05-03-24. The ESB 2021 is anonymous and confidential, in accordance with Law 6/2007, of 17 July, which regulates the preparation and publication of surveys and opinion polls in Catalonia. On the other hand, the confidentiality of the data is guaranteed in accordance with Organic Law 3/2018 on the Protection of Personal Data and the guarantee of digital rights. Therefore, it is assured that the information obtained from the questionnaires will be used exclusively in the field of health. According to the regulation, access to the data is on direct request to the administration (Public Health Agency of Barcelona) upon request by accredited research groups, under the clause of non-transfer to third parties. Therefore, access to the data is not allowed without their consent and subrogated access is not possible. On legal grounds, data is only accessible upon request from the official administrative source to the address info@aspb.cat.

**Funding:** The author(s) received no specific funding for this work.

**Competing interests:** None competing interests.

regeneration can encompass a broad range of policy sectors, from housing, public spaces and mobility, to social services, information technologies and economy. Moreover, social problems tend to be spatially distributed in an unequal way, which can make urban programmes more suitable than traditional sector programmes [3]. An unequal provision of resources among individuals and social groups in an urban environment can lead to health inequalities. Comprehensive urban regeneration programmes aim to facilitate access to and availability of a wide range of these resources. Therefore, it is expected that urban regeneration programmes comprising a greater diversity of interventions will favour access to dispositional resources and provide wider options for a better life. Finally, urban regeneration programmes can be seen as public health interventions that potentially improve the wellbeing of recipients and indirectly benefit the entire population [4, 5].

Previous literature reported that good built and social environments are associated with desirable social and health outcomes, like social cohesion and physical activity, and inversely associated with others like overweight and alcohol abuse [6, 7]. However, when individual characteristics are taken into consideration, the literature found mixed evidence for urban regeneration programmes to improve wellbeing [1, 8], especially for mental health [9–11]. The variability in results could be due to the heterogeneity of the interventions and different methodological threads. In particular: changes in population composition (e.g., gentrification), non-observed confounding factors, and previous trends; and different analyses (short- or long-term effects) [1, 12]. Despite variability, several studies found evidence for urban regeneration programmes to improve wellbeing in different situations: when citizens participate in them [13], in areas of high-intensity intervention [14–16], and among citizens residing in the intervention area for a long time [17].

Urban regeneration programmes have been implemented in Europe since the early 90's under the Urban pilot projects 1990–1993, and more decisively with the URBAN programmes (1994–2006). In this context, the Catalonia Neighbourhood's Law (Law 2/2004) was designed to improve living conditions in the most disadvantaged neighbourhoods, mainly through town planning interventions. In Barcelona city, these urban regeneration programmes have been associated with improvements in self-perceived health, mainly among low-income citizens [18]. Therefore, in 2016, the city council promoted a new urban regeneration programme, "Pla de Barris (PdB) 2016–2020". PdB was a multicomponent, comprehensive, intersectoral, and community engagement programme with a total budget of 150 million euros implemented in 16 disadvantaged neighbourhoods with a total population of 225,631 people in 2016. It aimed to decrease inequalities by involving citizens in the development of projects that addressed the built and social environments. It was assumed that the benefits of the intervention would be neighbourhood-wide, and not restricted to participants.

The objective of our study was to assess the impact of the PdB urban regeneration programme on wellbeing in deprived neighbourhoods of Barcelona, characterised by lower indicators in terms of income, education level, residential vulnerability indices, higher unemployment rates, a higher percentage of the foreign population and material deprivation. To achieve this, we assessed selected health outcomes and health-related behaviours through routine health survey data in a pre-post design with a comparison group. To overcome some methodological limitations mentioned above, a differentiation was made between the moderate and high-intensity intervention neighbourhoods according to the amount of investment per inhabitant. The difference-in-difference method with reweighting was applied to reduce possible bias in socio-economic characteristics between the comparison and intervention groups in the pre- and post-intervention periods.

## Materials and methods

### The intervention Pla de Barris (PdB)

The programme is a multicomponent, comprehensive intersectoral and community engagement programme aimed at improving living conditions in the most disadvantaged neighbourhoods. The total budget was of 150 million € (m€), of which 105 m€ were for inversion, 35 m€ were for services and 10 m€ were for overheads. The four components include: i) social rights: 277 actions with a budget of 27,8 m€ for the improvement of cohesion, promotion of culture and sports, attention to social services, housing and health, information technology and equipment; ii) education: 165 actions and 33,3 m€ devoted to schools and literacy education; iii) economy: 149 actions and 9,5 m€ in commerce, social economy and employment; iv) urban ecology: 122 actions and 70 m€ in public space, mobility and green areas.

It would be impossible to give an account of all of the 713 actions. The following are examples: intervention in schools by recruiting 160 social and health service professionals to reinforce the emotional education of pupils and families in the school environment; a childcare service was offered, mainly used by single-parent families, providing more than 11.2 thousand services; digital training courses were held; extending occupational training and developing disused ground floors for commercial purposes; bus lines were inaugurated to areas that are difficult to access; rehabilitation of vulnerable housing was carried out in defense of the right to decent housing; various adaptations of public space and the opening of green spaces with a gender perspective; the refurbishment and opening of open-access sports facilities. As 64 of the total 713 actions were strictly in the health sector, the assumption in this research is that the health benefits of an intervention in health determinants outside the health sector spill over to the entire population.

The participation of the neighbours has been sought to be carried out not only in the phases of diagnosis and definition of the objectives and actions but also during the follow-up and management.

### Design, study population and sources of information

This study has a quasi-experimental design, comparing the group of the 16 intervened neighbourhoods with a comparison group. Because of the variation in budget across neighbourhoods (335.8 to 2048.7 € per inhabitant), a distinction was made between moderate- and high-intensity interventions on the basis of the median of 727.6 € per inhabitant. Such distinction is also motivated by an interest in studying a possible dose-response effect.

The neighbourhoods of the comparison group were selected among the ones that were most similar to the intervened ones in terms of socioeconomic characteristics, such as household disposable income, percentage of people with primary education or less, unemployment, rate of people attending social services, and household overcrowding. With a Cronbach's alpha of 0.9, a summary index was computed with the first principal components. The 17 neighbourhoods with the lowest scores (apart from the ones in the intervention group) were selected to form the comparison group (with a population of 302,270 in 2016).

We used the data collected by the Public Health Agency of Barcelona and the city council from the Barcelona Health Surveys (BHS) of 2016 and 2021 (respectively, the first year of the intervention and the year after the end of it). The BHS is a routine official survey with the objective of understanding the health of the adult population (>14 years old) and its determinants that is planned every five years. As a small fraction of the intervention was carried out in 2016, we consider that there was insufficient time for any effects to have occurred. The sampling design of the BHS consists of 4000 randomly selected men and women distributed

among the 10 districts and 73 neighbourhoods of Barcelona. The sample size was computed to achieve a precision of ± 2% for the whole city and ±6% for the 10 districts. Interviews were conducted face-to-face by trained professionals using computer-assisted personal interviewing. Non-respondents are substituted by individuals in the same quotas (by sex, age group and neighbourhood) until the objective sample is reached. The final sample size for the 16 neighbourhoods in the intervention group and the 17 neighbourhoods in the control group was 558 men and 603 women in 2016 and 559 men and 601 women in 2021, which represents a precision of ± 3% for the 33 neighbourhoods analysed. All computations included sample weights to restore representativeness; moreover, potential bias among intervention and comparison groups in 2016 and 2021 was corrected by reweighting with the inverse of the propensity score.

## Variables

On the basis of previous literature, we focused on a range of health outcomes and health-related behaviours potentially related to urban regeneration programmes: psychosocial distress (mental health, use of psychotropic drugs, and alcohol abuse), self-perceived health status, physical activity, and obesity.

Mental health was measured with the General Health Questionnaire-12 (GHQ-12), which measures psychosocial distress with 12 items. Such items are added up to an index in the range 0–12 and then dichotomized: having a score of 3 or more means having poor or not-good mental health (reference category).

Use of psychotropic drugs was defined as having consumed at least 1 drug among antidepressants, tranquillizers, or sleeping pills in the last two days (reference category).

Alcohol abuse was measured according to the amount of alcohol consumed, the type of drink, and the frequency of consumption. Participants were categorized as risky drinkers or moderate/non-drinkers (reference category). Moreover, drinking five or more alcoholic beverages on a single occasion was considered risky consumption [19].

Self-perceived health status was reported in five categories and then dichotomized: "excellent", "very good", and "good" were classified as good health status (reference category), and "fair" and "poor" as poor health status.

Physical activity during leisure time was reported on the basis of the International Physical Activity Questionnaire (IPAQ). It was classified according to the energy required, its duration, and its frequency. Finally, it was dichotomized into moderate and intense activity and low-intensity activity (reference category) [19].

Obesity was evaluated on the basis of self-reported weight and height and categorised according to the cut-off points recommended by the WHO. Overweight, normal weight, and underweight were classified as non-obesity (reference category).

The propensity score was computed with the following covariates: sex; age group (15–34 as the reference category, 36–64, and 65 and above); occupational social class (manual as the reference category and non-manual); employment situation (employed, unemployed, and others); origin (autochthonous and foreign-born); and population turnover (having been living in the neighbourhood for 6 years or less, or having been living there for more than 6 years).

## Statistical analysis

To identify the potential effect of PdB on health outcomes and health-related behaviours, we used differences-indifferences (DiD) combined with a propensity score and an inverse probability weighting. DiD is a widely used method to assess the effectiveness of public health interventions and it represents a growing area of methodological development [20]. It consists in

deducing the effect of the intervention from the difference between the observed evolution after the intervention and the estimated evolution without the intervention. The latter is assumed to be equal to the observed evolution of the comparison group. In our study, we used a linear probability model to estimate the age-adjusted effect with 95% confidence intervals in percentages.

The attribution of causality in DiD is threatened by two factors: a non-parallel evolution between intervention and comparison groups prior to intervention and a different compositional evolution between the two groups. However, it has been suggested that these biases can be substantially reduced with a propensity score [21]. In our case, a probability was computed based on selected demographic and socioeconomic covariates (age, occupational social class, origin, and employment status). It was assumed that the selected covariates were not affected by the intervention. Then, a final score was computed with the inverse of the probability previously obtained along with the sampling weights.

Finally, the linear probability model estimation of the DiD incorporates these new weights. To account for individuals sharing neighbourhoods' characteristics, we applied clustered robust standard errors. All computations were stratified by gender. To check if results were maintained in case the comparison group changed, we performed a sensitivity analysis that included the 28 neighbourhoods of low- and middle-income.

## Population turnover

To assess compositional changes in the populations of the comparison and intervention neighbourhoods during the intervention, we compared the distribution by age group, sex, and educational level. Data were obtained from the official register of inhabitants from the city council for the years 2016 and 2021. We also checked differences in the number of years living in the neighbourhood before and after applying the propensity score.

## Results

The most frequent sociodemographic characteristics both of men and women in the intervention and comparison groups were the following: being 35- to 64-year-old, being employed in manual occupations, being autochthonous, and having been living for more than 6 years in the neighbourhood. In 2016, before the intervention, there was higher unemployment among men and more foreign-born men and women in the intervention groups (Table 1). After intervention, there were differences in age among women and in country of birth among men between the comparison and intervention groups. These differences were removed after weighting data derived from the propensity score for the pre-intervention and post-intervention period (S1 Table 1 in S1 File). Table 1 also shows no population turnover (gentrification) in the sample according to the number of years living in the neighbourhood. This was confirmed by the official population registration data: there were no differences by sex or age group between 2016 and 2021, and the education level improved at a common rate both in the comparison and the intervention groups (S1 Table 2 in S1 File).

Table 2 shows the results of the intervention on selected outcomes from the DiD weighted regression. High-intensity intervention significantly reduced the relative frequency of women with poor mental health by 15.5% with respect to the comparison group. In the high-intensity intervention group, the frequency of women with poor self-perceived health was significantly reduced by 21.7%, and the frequency of women doing moderate and intense physical activity significantly increased by 13.7%. For these two indicators, the improvement was mainly due to the favourable evolution of the high-intensity intervention group. Finally, the frequency of women with obesity significantly decreased (9.7%), but alcohol abuse increased (5.8%) in the

**Table 1. Characteristics of the study population stratified by sex and group.** Barcelona, 2016 and 2021.

| | 2016 | | | | | | 2021 | | | | | |
| | Men | | | Women | | | Men | | | Women | | |
| | Comparison N = 321 | Moderate intensity N = 177 | High intensity N = 60 | Comparison N = 348 | Moderate intensity N = 184 | High intensity N = 71 | Comparison N = 308 | Moderate intensity N = 199 | High intensity N = 52 | Comparison N = 351 | Moderate intensity N = 179 | High intensity N = 71 |
|---|---|---|---|---|---|---|---|---|---|---|---|---|
| **Age** | | | | | | | | | | | | |
| 15–34 | 25.1 | 31.1 | 22.9 | 26.0 | 27.0 | 18.3 | 25.9 | 36.4 | 27.5 | 24.8 | 33.0 | 34.0 |
| 35–64 | 54.3 | 51.9 | 51.1 | 45.2 | 49.8 | 58.8 | 54.2 | 51.6 | 54.9 | 49.3 | 44.0 | 54.5 |
| +65 | 20.6 | 17.0 | 26.0 | 28.8 | 23.2 | 22.9 | 19.9 | 12.0 | 17.6 | 25.9 | 23.0 | 11.5 |
| p-value[a] | 0.420 | | | 0.244 | | | 0.061 | | | **0.037** | | |
| **Occupational social class** | | | | | | | | | | | | |
| Non-manual | 16.1 | 12.0 | 7.4 | 15.1 | 11.9 | 14.2 | 20.7 | 22.1 | 13.6 | 22.6 | 19.6 | 15.2 |
| Manual | 81.5 | 85.6 | 85.1 | 83.0 | 84.7 | 84.3 | 77.9 | 74.7 | 86.4 | 75.7 | 75.9 | 82.1 |
| Missing | 2.4 | 2.4 | 7.5 | 1.9 | 3.4 | 1.5 | 1.4 | 3.2 | 0.0 | 1.7 | 4.5 | 2.7 |
| p-value | 0.100 | | | 0.677 | | | 0.367 | | | 0.366 | | |
| **Employment situation** | | | | | | | | | | | | |
| Employed | 53.2 | 55.8 | 45.5 | 45.1 | 49.9 | 45.3 | 55.4 | 57.1 | 55.0 | 49.9 | 49.4 | 54.7 |
| Unemployed | 9.0 | 16.8 | 18.2 | 7.9 | 10.0 | 13.6 | 11.1 | 15.0 | 14.0 | 9.6 | 10.2 | 8.8 |
| Others | 36.5 | 26.0 | 31.9 | 45.9 | 39.6 | 41.1 | 33.1 | 27.2 | 30.9 | 40.3 | 39.5 | 35.0 |
| Missing | 1.3 | 1.4 | 4.4 | 1.1 | 0.5 | 0.0 | 0.4 | 0.7 | 0.0 | 0.2 | 0.9 | 1.5 |
| p-value[a] | **0.020** | | | 0.531 | | | 0.591 | | | 0.935 | | |
| **Country of origin** | | | | | | | | | | | | |
| Autochthonous | 72.3 | 58.2 | 76.9 | 72.8 | 59.6 | 74.6 | 70.1 | 55.3 | 68.9 | 72.1 | 66.5 | 65.6 |
| Foreign born | 27.7 | 41.5 | 24.1 | 27.2 | 40.4 | 25.4 | 29.6 | 44.7 | 31.1 | 27.7 | 33.5 | 34.4 |
| Missing | 0 | 0.3 | 0.0 | 0 | 0 | 0 | 0.3 | 0.0 | 0.0 | 0.2 | 0.0 | 0.0 |
| p-value[a] | **0.005** | | | **0.006** | | | **0.002** | | | 0.285 | | |
| **Population turnover** | | | | | | | | | | | | |
| ≤6 years | 19.2 | 18.7 | 13.7 | 14.8 | 19.7 | 12.5 | 24.2 | 27.6 | 30.2 | 22.3 | 23.7 | 24.7 |
| >6 years | 80.4 | 80.0 | 84.9 | 85.2 | 80.3 | 87.5 | 75.4 | 72.4 | 69.8 | 77.7 | 75.3 | 75.3 |
| Missing | 0.4 | 1.3 | 1.4 | 0 | 0 | 0 | 0.4 | 0.0 | 0.0 | 0.0 | 1.0 | 0.0 |
| p-value[a] | 0.473 | | | 0.507 | | | 0.557 | | | 0.856 | | |

[a] in bold significance of p-values <0.05 of across comparison and intervention groups.

**Table 2. Health outcomes and health-related behaviours in the study population stratified by sex and group.** Barcelona, 2016 and 2021.

| | | Men | | | | Women | | | |
|---|---|---|---|---|---|---|---|---|---|
| | Type of intervention | Pre-2016[a] | Post-2021[a] | Post-Pre | DiD[b] (p-value) [c] | Pre-2016[a] | Post-2021[a] | Post-Pre | DiD[b] (p-value) [c] |
| **Poor mental health** | Comparison | 22.9 | 23.9 | 1.0 | - | 22.5 | 33.9 | 11.4 | - |
| | Moderate | 19.7 | 27.2 | 7.5 | 6.5 (0.472) | 31.0 | 33 | 2.0 | -9.4 (0.153) |
| | High | 17.2 | 22.6 | 5.4 | 4.4 (0.658) | 43.9 | 39.8 | -4.1 | **-15.5 (0.020)** |
| **Psychotropic drug use** | Comparison | 13.7 | 9.2 | -4.5 | - | 19.6 | 21.4 | 1.8 | - |
| | Moderate | 9.5 | 14.0 | 4.5 | 9.0 (0.057) | 24.0 | 24.7 | 0.7 | -1.1 (0.935) |
| | High | 6.0 | 11.8 | 5.8 | **10.3 (0.029)** | 24.9 | 14.1 | -10.8 | -12.6 (0.205) |
| **Alcohol abuse** | Comparison | 11.3 | 14.4 | 3.1 | - | 6.8 | 5.9 | -0.9 | - |
| | Moderate | 9.3 | 13.0 | 3.7 | 0.6 (0.924) | 4.0 | 8.9 | 4.9 | **5.8 (0.043)** |
| | High | 9.9 | 10.4 | 0.5 | -2.6 (0.691) | 6.0 | 6.3 | 0.3 | 1.2 (0.786) |
| **Poor health perception** | Comparison | 20.1 | 22.9 | 2.8 | - | 25.8 | 28.3 | 2.5 | - |
| | Moderate | 14.1 | 24.0 | 9.9 | 7.1 (0.311) | 26.8 | 27.4 | 0.6 | -1.9 (0.865) |
| | High | 20.0 | 22.4 | 2.4 | -0.4 (0.899) | 43.0 | 23.8 | -19.2 | **-21.7 (0.012)** |
| **Moderate and intense physical activity** | Comparison | 34.0 | 31.7 | -2.3 | - | 24.1 | 21 | -3.1 | - |
| | Moderate | 33.9 | 33.6 | -0.3 | 2.0 (0.865) | 15.3 | 18.8 | 3.5 | 6.6 (0.213) |
| | High | 31.7 | 33.7 | 2.0 | 4.3 (0.726) | 7.5 | 18.1 | 10.6 | **13.7 (0.025)** |
| **Obesity** | Comparison | 16.1 | 18.8 | 2.7 | - | 14.6 | 21.2 | 6.6 | - |
| | Moderate | 19.9 | 21.1 | 1.2 | -1.5 (0.834) | 21.0 | 17.9 | -3.1 | **-9.7 (0.022)** |
| | High | 17.2 | 12.4 | -4.8 | -7.5 (0.329) | 20.9 | 25.7 | 4.8 | -1.8 (0.706) |

[a]Prevalences/100.

[b]difference-in-difference estimated by linear regressions adjusted by age and weighted data by the inverse of the propensity score.

[c]in bold significance of p-value<0.05.

moderate-intensity intervention group; however, these results were not significant in the sensitivity analysis (S1 Table 3 in S1 File).

Regarding men, the only significant result was an increase of 10.3% in the frequency of those using psychotropic drugs in the high-intensity intervention group. This result was due to a drop in the comparison group, while the two intervention groups showed a parallel ascending trend.

## Discussion

This study found a positive impact of urban regeneration programmes on mental health, self-perceived health, and physical activity among women in the high-intensity intervention group.

Apart from the amount invested, citizen participation also seemed to favour the effectiveness of our intervention [13, 15, 16]. The improvements in women's health but not in men's may be related to many factors. First, women participated more than men in the actions of the PdB. Indeed, women in the most deprived neighbourhoods are usually less included in the labour market and so tend to have more free time. In addition, in order to reconcile work and family life, female workers often choose jobs close to home or part-time [22]; therefore, they

might spend more time in their neighbourhoods and be more sensitive to the impact of an urban regeneration programme than men [23]. Moreover, women in the intervention groups showed worse health outcomes at pre-intervention than men, leaving more room for improvement.

An improvement in mental health was previously detected among different populations in other urban regeneration programmes: among women of the high-intensity intervention group in the neighbourhood-wide intervention 'District Approach' in the Netherlands [23]; among men and women in the programme 'GoWell' in Glasgow [14]; in deprived areas in Wales, without differentiation by gender [17]. Moreover, in our study, indicators related to psychological distress and physical health status move in the same favourable direction among women. In the Spanish context, previous research has also found effects of benefits (on preventable causes of mortality) in areas of higher investment intensity where two or more urban regeneration programmes overlap [24, 25]. In general, although wide-neighbourhood intervention programmes and citizen participation seem to benefit mental health in areas of high-intensity intervention, the evidence is not clear. For instance, the evaluation of the participatory 'New Deal for Communities' intervention in England found no overall effect on mental health, except for most disadvantaged groups [26], neither in the area of intervention in Glasgow [2], nor in the urban regeneration in Northern Ireland [27], nor in the 'Well London' project [14].

Self-perceived health status is a less studied outcome. There is weak evidence for its improvement in the 'Neighbourhood Law' for 2006–2011 in Barcelona [18] and in the neighbourhood renewal programme in Northern Ireland [25]. No effect on self-perceived health status was observed neither in the greening space intervention in the context of the 'District Approach' in the Netherlands [28], nor in the 'Neighbourhood Renewal Strategy' in Australia, except for the small subgroup involved in the partnership activities [29]. In our case, the reduction in poor self-perceived health among women is consistent with the reduction in poor mental health; both pre-intervention frequencies evolved downward.

The effects of the interventions on physical activity have been shown to depend on citizen participation [30], and the quality, safety, and civility of urban green spaces [31]. Even though our evaluation makes no distinction between actions undertaken for these different areas of action, we observed an overall improvement in physical activity among women in the high-intensity intervention group. In fact, physical activity and obesity move in the same direction in favour of the intervention for both sexes, even though they are not all statistically significant. In our case, as in a previous study, more walkable public spaces may have been the cause of the improvement in overall health [32].

The increase in the use of psychotropic drugs by men in the high-intensity group was an unexpected result; however, it was driven by a drop in the comparison group. This result contrasts with the null effects found in Denmark for the years 2015 to 2020 [33], and with the favourable evolution of residents in urban regeneration areas in Andalusia (Spain) during the years 2008 to 2015 [34]. We must bear in mind that during the years under study, there have also been two relevant circumstances. On the one hand, the long-lasting effects of the economic crisis during 2016 and the emergence of the COVID-19 pandemic in early 2020. Both events have led to a worsening of mental-related health problems, especially among the male population during the economic crisis and among young adult women during the lockdown of COVID-19. It is difficult to identify to what extent both crises may have differentially affected our comparison and intervention groups and through which mechanisms. If this were the case, it would be a limitation of our research. There is evidence of oscillations in health inequalities during both crises, but in the medium term, they translate into the persistence of inequalities [35, 36]. For this reason, we believe that the effects of both crises on the different

evolution of health status between the intervention and control groups are small. In any case, the effects of the intervention must be interpreted depending on the contextual characteristics of the neighbourhoods.

Of the lessons learned about intervention design, the excessive number of interventions rather than focusing on the most determinant ones in each area of action is noteworthy. Second, participation is uneven across neighbourhoods and is lower in the most deprived neighbourhoods. It is therefore necessary to implement a plan to strengthen the capacity for collective action in these neighbourhoods.

This study has several limitations. Caution should be exercised when interpreting the results causally. Even though we used comparison areas and propensity score to make the distributions of population features between comparison and intervention groups comparable, threads of causality could arise if other factors affect both groups differently in previous trends or during the intervention. If the health indicators between the intervention and control groups did not move in parallel in the previous periods, this would imply that factors other than the intervention were at work and the results obtained could not be attributed to the intervention. For example, if the lasting effects of the 2008 economic crisis had affected the intervention group more severely than the control group. This could have further worsened their health indicators in 2016. In such a case, these indicators would tend to return to normal in 2021, and the improvement would not be attributable to the intervention. Another circumstance that could occur is that the intervention itself has changed the social context, and therefore the observed improvements in health cannot be attributed to the intervention. However, we believe that these effects, even if they exist, should be limited and not compromise the overall results. A low sample size is another limitation, especially in the high-intervention group, which does not allow for the analysis of the impact on health inequalities or the testing of heterogeneous results among socioeconomic groups within neighbourhoods. It was not possible to assess if the effect among participants was greater because of the lack of data. Finally, the time when to study the impact of an intervention represents a controversial issue: on the one hand, it may take time to change health behaviours, such as dietary habits or obesity; on the other hand, the impact of the interventions may decay and confounding factors may increase with time. For example, limited results were found in the 'New Deal for Communities' intervention within 3.5 years after the intervention, but no results were found after 6 years [26], nor in the 'District Approach' intervention [37]. Therefore, for our intervention, a long-term evaluation will be needed to see if the positive health outcomes are sustained over time.

This study also has several strengths. First, it adds evidence to the impact of regeneration programmes on health outcomes and health-related behaviours, an area where evidence is still scarce. Second, the different surveys make it possible to have numerous comparable health outcomes in the two periods and will allow a long-term evaluation. Third, the quasi-experimental design with a comparison group helps to minimise the threats to the internal validity of the study. Finally, the analysis used propensity score to reduce potential biases from possible failure to meet the parallel assumption and diminish the potential compositional change between comparison and intervention groups during the period.

Our results can be useful to implement urban regeneration programmes in other areas. In particular, we derived several lessons from them. First, urban planning practice can probably do better to integrate insights from public health, explicitly establishing the links between the built environment and health outcomes. Second, under budgetary constraints, actions should be designed to concentrate resources on the target population or areas with worse living conditions within neighbourhoods where there is more room for improvement. Third, a specific strategy to target men is needed. Fourth, a qualitative study to report on the perceptions and experiences of local residents could play a pivotal role in strengthening the study's ability to

establish a more robust causal link between the regeneration programme and the observed positive health outcomes. Finally, the re-assessment of urban regeneration programmes at different times would make it possible to disentangle if the impact observed is maintained in the long term.

## Conclusions

This study shows positive short-term effects of the urban regeneration programme Pla de Barris 2016–2020 on health outcomes in women in the high-intensity intervention group. These results can guide future interventions in other areas.

## Supporting information

**S1 File.**
(DOCX)

## Author Contributions

**Conceptualization:** Xavier Bartoll-Roca, María José López, Katherine Pérez, Lucía Artazcoz, Carme Borrell.

**Data curation:** Xavier Bartoll-Roca.

**Formal analysis:** Xavier Bartoll-Roca.

**Methodology:** Xavier Bartoll-Roca.

**Supervision:** Xavier Bartoll-Roca.

**Writing – original draft:** Xavier Bartoll-Roca.

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
