## [Decision Letter · Decision Letter 0]

4 Dec 2023

PONE-D-23-31700Short-term health effects of an urban regeneration programme in deprived neighbourhoods of BarcelonaPLOS ONE

Dear Dr. Bartoll,

Thank you for submitting your manuscript to PLOS ONE. After careful consideration, we feel that it has merit but does not fully meet PLOS ONE’s publication criteria as it currently stands. Therefore, we invite you to submit a revised version of the manuscript that addresses the points raised during the review process.

**ACADEMIC EDITOR:** In light of both reviewers opinion and considering their comments, the article needs to address various aspects, from survey/data limitations to the sound discussion of results. It is strongly recommended to address all comments and questions raised in order to be accepted for publication, as it was considered a very important study.

We look forward to receiving your revised manuscript.

Kind regards,

Angela Mendes Freitas

Academic Editor

PLOS ONE

Journal Requirements:

"None competing interests"

4. Please amend the manuscript submission data (via Edit Submission) to include authors Dr. Maria José López, Dr. Katherine Pérez, Dr. Lucia Artazcoz and Dr. Carme Borrell.

Reviewers' comments:

Reviewer's Responses to Questions

**Comments to the Author**

1. Is the manuscript technically sound, and do the data support the conclusions?

Reviewer #1: Yes

Reviewer #2: Partly

Reviewer #3: Yes

2. Has the statistical analysis been performed appropriately and rigorously? 

Reviewer #1: Yes

Reviewer #2: Yes

Reviewer #3: Yes

3. Have the authors made all data underlying the findings in their manuscript fully available?

Reviewer #1: Yes

Reviewer #2: No

Reviewer #3: No

4. Is the manuscript presented in an intelligible fashion and written in standard English?

Reviewer #1: Yes

Reviewer #2: No

Reviewer #3: Yes

5. Review Comments to the Author

Reviewer #1: An exhaustive description on the renewable solutions adopted is needed

1. The study presents the results of original research. Yes

2. Results reported have not been published elsewhere. Yes

3. Experiments, statistics, and other analyses are performed to a high technical standard and are described in sufficient detail. Yes

4. Conclusions are presented in an appropriate fashion and are supported by the data. Yes

5. The article is presented in an intelligible fashion and is written in standard English. Yes

6. The research meets all applicable standards for the ethics of experimentation and research integrity. Yes

7. The article adheres to appropriate reporting guidelines and community standards for data availability. Yes

Reviewer #2: This paper reports on the impact of an urban regeneration programme in Barcelona. The results confirm results of previous studies in other countries of a modest impact among women in particular. I do have multiple questions, however, relating to the validity and interpretation of the results:

1 . I could not find information on non-respons on the survey. It seems that 4000 people have been approached (per sexe?), and around 600 participated. Right? This implies a response rate of 15%, which is extremely low. I would like to see information on the selectiveness of the response, as this might bias the results.

2. I miss information on the socio-economic characteristics at neighbourhoodlevel. What is the level of deprivation here? This information is also useful in relation to my first point: is the survey representative for the population at stake?

3. What exactly does the programme entail? The authors state that they could not describe all activities, which I of course do understand. As a reader, however, I have no clue as to what the content of the programme was. E.g. classified as ‘economy’: how substantial have these been? How many people have been reached. And did the programme also entail housing renovations for example? If policymakers want to base their decisions on studies like this, they need information as what activities/programmes are needed to achieve this result.

4. Related to the previous question: what exactly was the budget per neighbourhood? Is it 28 million in total, distributed across 16 neighbourhoods, over 4 years? It seems a relatively small amount per neighbourhood, also depending of course on how many people live in a neighbourhood, what determines the amount of money spent per person. A similar question relates to the difference between moderate and high intensity districts. Is this a relevant difference?

5. The authors emphasize the positive results among women in high intensity districts. However, 2 of the positive effects occur in low intensity districts. So is there really a difference?

6. Some of the results seem odd, and need to be discussed. E.g.

-reduction in obesity among women in moderate intensity disricts, as compared to increase in high intensity districs.

-the prevalence of poor health perception in women in high intensity district is extremely high (43%), which could probably explain the observed decrease.

-the observed increase in drug use among men in intervention districts.

7. An important weakness is the absence of trend data. The authors use two time points, which might bias the results. This should at least be mentioned in the discussion section.

Reviewer #3: The manuscript entitles "Short-term health effects of an urban regeneration programme in deprived neighborhoods of Barcelona" is interesting and makes a significant contribution to this research field in Spain. However, I believe that before being accepted for publication, the authors should address a series of revisions that I consider important:

1. In the introduction, the authors make the following statement regarding the spatial distribution of social problems and the suitability of urban regeneration projects in neighborhoods with a high density of immigrant population: "especially in immigrant-dense neighbourhoods, which can make urban programmes more suitable than traditional sector programmes." I believe that this assumption deserves a better argumentative development, an explanation of why these types of problems can be more effectively addressed through urban regeneration.

2. The researchers conduct a limited review of the existing literature and overlook the consideration of previous studies published in Spain on the subject. For example, they omit studies that specifically illustrated potential effects in areas of high intervention intensity (confluence of two or more project-based areas).

Moya, A. R. Z., & Yáñez, C. J. N. (2017). Impact of area regeneration policies: Performing integral interventions, changing opportunity structures and reducing health inequalities. J Epidemiol Community Health, 71(3), 239-247

Even the existence of relevant prospective studies published is dismissed by stating that "there is a lack of longitudinal studies.

Rodgers, S. E., Heaven, M., Lacey, A., Poortinga, W., Dunstan, F. D., Jones, K. H., ... & Lyons, R. A. (2014). Cohort profile: the housing regeneration and health study. International journal of epidemiology, 43(1), 52-60.

Smith, N. R., Clark, C., Fahy, A. E., Tharmaratnam, V., Lewis, D. J., Thompson, C., ... & Cummins, S. (2012). The Olympic Regeneration in East London (ORiEL) study: protocol for a prospective controlled quasi-experiment to evaluate the impact of urban regeneration on young people and their families. BMJ open, 2(4), e001840.

Nygaard, S. S., Jorgensen, T. S. H., Wium-Andersen, I. K., Brønnum-Hansen, H., & Lund, R. (2023). Is urban regeneration associated with antidepressants or sedative medication users: a registry-based natural experiment. J Epidemiol Community Health, 77(4), 237-243.

Cummins, S., Clark, C., Lewis, D., Smith, N., Thompson, C., Smuk, M., ... & Eldridge, S. (2018). The effects of the London 2012 Olympics and related urban regeneration on physical and mental health: the ORiEL mixed-methods evaluation of a natural experiment. Public Health Research, 6(12).

Zapata-Moya, Á. R., Martín-Díaz, M. J., & Viciana-Fernández, F. J. (2021). Area-Based Policies and Potential Health Benefits: A Quasi-Experimental Cohort Study in Vulnerable Urban Areas of Andalusia (Spain). Sustainability, 13(15), 8169.

3. In the introduction, the authors refer to the lack of prospective studies as a factor that raises a number of weaknesses in studies on the health impact of urban regeneration programs, and mention specifically the following (Changes in the composition of the population; Failure to consider confounding factors and Previous trends). Although the methodological effort made by the researchers to develop a quasi-experimental approach is commendable, the cross-sectional nature of the data used does not allow in any case to overcome the three limitations mentioned above, especially the risk of attributing effects to the intervention without controlling for the population change that occurred before and after the intervention, as well as the study of previous trends in the most vulnerable neighborhoods. Combining the quasi-experimental method with the individual-level propensity score technique is an alternative to try to reduce these risks; however, the specialized literature indicates that there is a greater population change in neighborhoods intervened by urban regeneration programs. The researchers try to show that these differences in population change are not operating in their analyses, but I believe they should acknowledge that having only two cross-sectional sampling points does not allow for a categorical assertion of the non-existence of effects due to the population change motivated by the interventions.

4. In addition to the limitations mentioned, there are external factors (the 2008 economic crisis and subsequent recovery, which some studies place from 2016 onward) that could be affecting the results, especially in those more vulnerable neighborhoods, as there could be a r"regression to the mean effect" after a greater worsening of the main outcomes in the crisis phase. This could be happening especially among women, as the authors indicate; these showed worse health outcomes than men before the intervention, and therefore, there would be more possibility of a tendency (beyond the intervention) to improve, a "regression to the mean effect" among a particularly vulnerable group in a context of vulnerability. To rule out this alternative explanation for the main findings of the study, trends in these indicators should be explored jointly in intervened and control neighborhoods whenever possible.

5. It is also important to mention another external factor that could influence the results in some way: since the COVID-19 crisis and lockdown occurred in 2020, the researchers should at least comment on how this could be impacting the potential effects found on mental health, self-perception of health among women, and alcohol consumption. Similarly, with the increase in the use of psychotropic drugs among men.

6. Regarding these external factors, the researchers in the discussion section point out as a strength of the study that "Indeed, any other contextual factor affecting the results at the city level should be equally affecting all the groups studied." In my opinion, this is a risky statement, as there is no guarantee that the impact of the economic crisis, as well as its recovery period, and even the COVID-19 crisis (as external factors) will have a similar effect among all city groups and all neighborhoods. Also, the potential impacts of the programs on health could be conditioned by the differential effects of the economic crisis.

Zapata Moya, Á. R., & Navarro Yáñez, C. J. (2021). Urban regeneration policies and mental health in a context of economic crisis in Andalusia (Spain). Journal of Housing and the Built Environment, 36, 393-405.

7. In a more nuanced interpretation, I think it is essential to recognize, especially in the discussion of the article, the potential for conditional effects of the intervention depending on the contextual characteristics of the neighborhoods.

8. Regarding mental health, the researchers choose to dichotomize the scale variable from a score of 3, considering that from this score onwards, there is a risk to mental health. I believe that this necessitates a more detailed explanation and references from reputable sources to support the authors' decision.

9. The explanation for why only positive effects are observed among women refers to a conventional argument that rests on the idea that women have greater exposure to community life in the neighborhood and, therefore, to the interventions of these projects. Perhaps the discussion could be enriched if the authors delve deeper into this matter and are able to reference various explanatory pathways (which may be complementary) from the perspective of the sociology of health.

Regarding the style of presenting information and the article's wording, I find it suitable for publication after addressing the points and suggestions made earlier.

6. PLOS authors have the option to publish the peer review history of their article (what does this mean?). If published, this will include your full peer review and any attached files.

Reviewer #1: No

Reviewer #2: No

Reviewer #3: No

---

## [Author Response · Author response to Decision Letter 0]

4 Feb 2024

PONE-D-23-31700

Short-term health effects of an urban regeneration programme in deprived neighbourhoods of Barcelona

We are grateful for the opportunity to address the issues raised by the reviewers, which undoubtedly improve the content of the article. In the following text we answer the reviewers' comments point by point. Corrections and modifications have been incorporated into the main manuscript in red and supressed text in change control.

Reviewers' comments and author’s responses:

Comments to the Author

1. Is the manuscript technically sound, and do the data support the conclusions?

Reviewer #1: Yes

Reviewer #2: Partly

Reviewer #3: Yes

RESPONSE:

In the following we comment on the changes made in accordance with the reviewers' recommended changes.________________________________________

2. Has the statistical analysis been performed appropriately and rigorously? 

Reviewer #1: Yes

Reviewer #2: Yes

Reviewer #3: Yes

3. Have the authors made all data underlying the findings in their manuscript fully available?

The requires authors to make all data underlying the findings described in their manuscript fully available without restriction, with rare exception (please refer to the Data Availability Statement in the manuscript PDF file). The data should be provided as part of the manuscript or its supporting information, or deposited to a public repository. For example, in addition to summary statistics, the data points behind means, medians and variance measures should be available. If there are restrictions on publicly sharing data—e.g. participant privacy or use of data from a third party—those must be specified.

Reviewer #1: Yes

Reviewer #2: No

Reviewer #3: No

RESPONSE:

We have extended the explanation of data availability in the cover letter. (There are legal restrictions)

The Barcelona Health Survey (BHS) forms part of the statistical actions of interest to the Generalitat de Catalunya and is included in the Annual Statistical Action Programme (PAAE) under the registration number: 05-03-24. The BHS 2021 is anonymous and confidential, in accordance with Law 6/2007, of 17 July, which regulates the preparation and publication of surveys and opinion polls in Catalonia. On the other hand, the confidentiality of the data is guaranteed in accordance with Organic Law 3/2018 on the Protection of Personal Data and the guarantee of digital rights. Therefore, it is assured that the information obtained from the questionnaires will be used exclusively in the field of health. According to the regulation, access to the data is on direct request to the administration (Public Health Agency of Barcelona) upon request by accredited research groups, under the clause of non-transfer to third parties. Therefore, access to the data is not allowed without their consent and subrogated access is not possible. Data is only accessible upon request from the official administrative source to the Public Health Agency of Barcelona address info@aspb.cat.

4. Is the manuscript presented in an intelligible fashion and written in standard English?

Reviewer #1: Yes

Reviewer #2: No

Reviewer #3: Yes

RESPONSE:

A second revision of the manuscript has been carried out by an expert translator.

5. Review Comments to the Author

Reviewer #1: An exhaustive description on the renewable solutions adopted is needed

1. The study presents the results of original research. Yes

2. Results reported have not been published elsewhere. Yes

3. Experiments, statistics, and other analyses are performed to a high technical standard and are described in sufficient detail. Yes

4. Conclusions are presented in an appropriate fashion and are supported by the data. Yes

5. The article is presented in an intelligible fashion and is written in standard English. Yes

6. The research meets all applicable standards for the ethics of experimentation and research integrity. Yes

7. The article adheres to appropriate reporting guidelines and community standards for data availability. Yes

Reviewer #2: This paper reports on the impact of an urban regeneration programme in Barcelona. The results confirm results of previous studies in other countries of a modest impact among women in particular. I do have multiple questions, however, relating to the validity and interpretation of the results:

1 . I could not find information on non-respons on the survey. It seems that 4000 people have been approached (per sexe?), and around 600 participated. Right? This implies a response rate of 15%, which is extremely low. I would like to see information on the selectiveness of the response, as this might bias the results.

We have introduced changes that hopes it makes this issue easier to follow. We have added information that the 4000 people interviewed from the 10 Districts refer also to the 73 neighbourhoods of Barcelona. We added information on quotes: by sex, age group and neighbourhood. As stated in the main text, non-response is supplemented by interviewing substitutes with the same characteristics until the target sample of 4000 cases for the whole Barcelona is reached, the resulting sample after the substitutions is comparable to the original sample design. Therefore, the response rate it is not 15%, there is no non-response. We have made it clearer that the analysed sample (from the 4000 cases for the whole BCN) corresponds to the subsample of the 16 neighbourhoods of the intervention group and the 17 neighbourhoods of the control group, which represents a precision of ± 3% for the 33 neighbourhoods analysed.

2. I miss information on the socio-economic characteristics at neighbourhood level. What is the level of deprivation here? This information is also useful in relation to my first point: is the survey representative for the population at stake?

We have introduced a paragraph detailing the characteristics of these vulnerable neighbourhoods: 

“[These neighbourhoods are] … characterised by poor indicators in terms of income, education level, residential vulnerability indices and higher unemployment rates, higher percentage of the foreign population and material deprivation”

We also have added information in the Methods section on the accuracy of estimates. The subsample for the 33 neighbourhoods analysed is representative with a precision of ± 3%. 

3. What exactly does the programme entail? The authors state that they could not describe all activities, which I of course do understand. As a reader, however, I have no clue as to what the content of the programme was. E.g. classified as ‘economy’: how substantial have these been? How many people have been reached. And did the programme also entail housing renovations for example? If policymakers want to base their decisions on studies like this, they need information as what activities/programmes are needed to achieve this result.

In the introduction there is a general idea regarding the aim of the urban regeneration programmes. We have added a summary sentence about the general goal of the intervention at the beginning of the intervention comment “… aimed at improving living conditions in the most disadvantaged neighbourhoods”.

Also, we give examples of relevant type of intervention. As examples of economic interventions, we already included increase the employability through “extending occupational training” or improving economic environment though “developing disused ground floors for commercial purposes”. Regarding housing rehabilitation, we already make explicit “rehabilitation of vulnerable housing was carried out”.

More information on the programme can be found on the web on some links below. However, we are reluctant to cite in the text because these links easily can change and become obsolete. Moreover, as the intervention progress, the information on these links is renewed and become obsolete easily. If it is a real problem we could add the general information link: https://www.pladebarris.barcelona/

Other links are:

Current population assessment: https://www.barcelona.cat/infobarcelona/en/tema/city-council/the-neighbourhood-plan-has-a-positive-impact-on-65-of-residents-in-23-neighbourhoods_1266510.html

Regarding housing interventions: https://www.barcelona.cat/infobarcelona/en/tema/city-council/the-neighbourhood-plan-allocates-over-21-million-euros-to-decent-housing-initiatives_920128.html).

More information about the specific activities related to the COVID-19 pandemics: (https://www.barcelona.cat/infobarcelona/en/tema/city-council/the-neighbourhood-plan-has-a-positive-impact-on-65-of-residents-in-23-neighbourhoods_1266510.html)

Regarding the intervention as accomplishing good practice by the city council (https://www.coe.int/eu/web/interculturalcities/-/barcelona-s-neighbourhood-plan), 

4. Related to the previous question: what exactly was the budget per neighbourhood? Is it 28 million in total, distributed across 16 neighbourhoods, over 4 years? It seems a relatively small amount per neighbourhood, also depending of course on how many people live in a neighbourhood, what determines the amount of money spent per person. A similar question relates to the difference between moderate and high intensity districts. Is this a relevant difference?

Thanks for the question. More details on the implementation can be found in the Catalan language document https://www.pladebarris.barcelona/, is not English-translated and difficult to navigate so not worth to refer to in the main text.

The paper already states that the total amount is 150 million €, of which 105 m€ were for inversion, 35 m€ were for services and 10 m€ were for overheads. We have moved this sentence to the beginning of the paragraph to avoid misunderstanding.

We have shown the total expenditure per capita in the neighbourhoods because we believe it is a good measure for the reader, and easily comparable with other common indicators such as income per capita or salary. So that, in the main text it is explicit the amount of investment in euros by inhabitant in the range of “335.8 to 2048.7 € per inhabitant”, which is not a negligible amount by Spanish standards, considering also that it comes from a local administration. Precisely, given the wide range between these magnitudes makes sense the distinction by intensity in €/hab. More resources were allocated to the most depressed areas in need of greater investment, hence the differentiation into high intensity and moderate intensity interventions. 

5. The authors emphasize the positive results among women in high intensity districts. However, 2 of the positive effects occur in low intensity districts. So is there really a difference?

We definitely believe that it is very convenient to keep the difference. One reason is conceptual: to allow ex-ante for intensity/dose effect. The second, is that previous literature that found results make the differentiation, this literature is commented in the discussion. Third, a very consistent association is observed for women (5 out of the 6 results analysed while only 1 for households). 

6. Some of the results seem odd, and need to be discussed. E.g.

-reduction in obesity among women in moderate intensity disricts, as compared to increase in high intensity districs.

-the prevalence of poor health perception in women in high intensity district is extremely high (43%), which could probably explain the observed decrease.

-the observed increase in drug use among men in intervention districts.

We also have stated the initial worse level of health status among women and interpreted as there are more room for improvement as in here: “Moreover, women in the intervention groups showed worse health outcomes at pre-intervention than men, leaving more room for improvement”. 

Following the reviewer suggestion, we have added as a limitation the possible case of ‘regression to mean’: “For example, if the lasting effects of the 2008 economic crisis had affected the intervention group more severely than the control group, this could have further worsened their health indicators in 2016. In such a case, these indicators would tend to return to normal in 2021 and the improvement would not be attributable to the intervention.”

We have also extended the comments regarding the increase in drug use among men: “This result contrasts with the null effects found in Denmark for the years 2015 to 2020 (Nygaard 2023), and with the favourable evolution of residents in urban regeneration areas in Andalusia (Spain) during the years 2008 to 2015 (Zapata-Moya 2021). We must bear in mind that during the years under study, there have also been two relevant circumstances. On the one hand, the long-lasting effects of the economic crisis during 2016 and the emergence of the COVID-19 pandemic in early 2020. Both events have led to a worsening of mental-related health problems, especially among the male population during the economic crisis and among young adult women during the lockdown of COVID-19. There is evidence of oscillations in health inequalities during both crises, but in the medium term, they translate into the persistence of inequalities35 36. For this reason, we believe that the effects of both crises on the different evolution of health status between the intervention and control groups are small. In any case, the effects of the intervention must be interpreted depending on the contextual characteristics of the neighbourhoods.” 

There is a decrease in obesity in all intervention groups for men and women compared to the control group, although only statistically significant for the moderate intensity group for women, which runs in favour of the intervention. This fact is congruent with the also increase in all interviewed group in physical activity. We have added: “In fact, physical activity and obesity move in the same direction in favor of the intervened groups of both sexes, even though they are not all statistically significant.”

7. An important weakness is the absence of trend data. The authors use two time points, which might bias the results. This should at least be mentioned in the discussion section.

We see the point of the reviewer. The BHS survey is routinely carried out every 5 years. Although we had a previous one for 2011, we decided not to include in this analysis to avoid introducing more noise into the data, as at this time it was affected by the financial crisis. However, we were aware that there may be trend problems (non-compliance with the hypothesis of parallel trends). For this reason, we have used the reweighting techniques to reduce possible biases between the intervention and comparison group before and after the intervention, as stated in the methodological section. 

Following the reviewer recommendation, we have expanded on this and other related issues in the limitations section: “This study has several limitations. Caution should be exercised when interpreting the results causally. Even though we used comparison areas and propensity score to make the distributions of population features between comparison and intervention groups comparable, threads to causality could arise if other factors differently affect both groups in previous trends or during the intervention. If the health indicators between the intervention and control groups do not move in parallel in the previous periods. This would imply that factors other than the intervention are at work and the results obtained could not be attributed to the intervention. For example, if the lasting effects of the 2008 economic crisis had affected the intervention group more severely than the control group, this could have further worsened their health indicators in 2016. In such a case, these indicators would tend to return to normal in 2021 and the improvement would not be attributable to the intervention, which is known as regression to the mean. Another circumstance that could occur is that the intervention itself has changed the social context and therefore these observed improvements in health cannot be attributed to the intervention. However, we believe that these effects, even if they exist, should be limited and do not compromise the overall results.” 

 Reviewer #3: The manuscript entitles "Short-term health effects of an urban regeneration programme in deprived neighbourhoods of Barcelona" is interesting and makes a significant contribution to this research field in Spain. However, I believe that before being accepted for publication, the authors should address a series of revisions that I consider important:

1. In the introduction, the authors make the following statement regarding the spatial distribution of social problems and the suitability of urban regeneration projects in neighbourhoods with a high density of immigrant population: "especially in immigrant-dense neighbourhoods, which can make urban programmes more suitable than traditional sector programmes." I believe that this assumption deserves a better argumentative development, an explanation of why these types of problems can be more effectively addressed through urban regeneration.

Sector programmes only account for a dimension of the problem. –we are not referring on urban regeneration programs but integral urban regeneration programs which are designed to account for a wide range of problems, acting on not only physical environment but on the social environment with the aim to redistribution of territorial, social and territorial endowments (which has been conceptualised as the Fundamental Cause Theory (Link, B.G.; Phelan, J. Social conditions as fundamental causes of disease. J. Health Soc. Behav. 1995, 35, 80–94). 

We make it explicit adding the following paragraph: “An unequal provision of resources among individuals and social groups in an urban environment can lead to health inequalities. Comprehensive urban regeneration programmes aim to facilitate access to and availability of a wide range of these resources. Therefore, it is expected that urban regeneration programmes comprising a greater diversity of interventions will favour access to dispositional resources and provide wider options for a better life”.

The reference to the immigrant population was not necessary and has been supressed.

2. The researchers conduct a limited review of the existing literature and overlook the consideration of previous studies published in Spain on the subject. For example, they omit studies that specifically illustrated potential effects in areas of high intervention intensity (confluence of two or more project-based areas).

Moya, A. R. Z., & Yáñez, C. J. N. (2017). Impact of area regeneration policies: Performing integral interventions, changing opportunity structures and reducing health inequalities. J Epidemiol Community Health, 71(3), 239-247

Even the existence of relevant prospective studies published is dismissed by stating that "there is a lack of longitudinal studies.

Rodgers, S. E., Heaven, M., Lacey, A., Poortinga, W., Dunstan, F. D., Jones, K. H., ... & Lyons, R. A. (2014). Cohort profile: the housing regeneration and health study. International journal of epidemiology, 43(1), 52-60.

Nygaard, S. S., Jorgensen, T. S. H., Wium-Andersen, I. K., Brønnum-Hansen, H., & Lund, R. (2023). Is urban regeneration associated with antidepressants or sedative medication users: a registry-based natural experiment. J Epidemiol Community Health, 77(4), 237-243.

Smith, N. R., Clark, C., Fahy, A. E., Tharmaratnam, V., Lewis, D. J., Thompson, C., ... & Cummins, S. (2012). The Olympic Regeneration in East London (ORiEL) study: protocol for a prospective controlled quasi-experiment to evaluate the impact of urban regeneration on young people and their families. BMJ open, 2(4), e001840.

Cummins, S., Clark, C., Lewis, D., Smith, N., Thompson, C., Smuk, M., ... & Eldridge, S. (2018). The effects of the London 2012 Olympics and related urban regeneration on physical and mental health: the ORiEL mixed-methods evaluation of a natural experiment. Public Health Research, 6(12).

Zapata-Moya, Á. R., Martín-Díaz, M. J., & Viciana-Fernández, F. J. (2021). Area-Based Policies and Potential Health Benefits: A Quasi-Experimental Cohort Study in Vulnerable Urban Areas of Andalusia (Spain). Sustainability, 13(15), 8169.

Thank you for noticing the issue and facilitating these important references. The statement "there is a lack of longitudinal studies” has been supressed; it was mean to say that retrospective studies are more common than prospective ones.

We have cited two papers from Moya & Yáñez, (2017) and Zapata-Moya 2021, precisely to reinforce the importance of integrated urban regeneration programs instead of sector programmes (even though it deals with mortality) and the paper from Nygaard to add (lack of) evidence on drug use.

We added the following texts in the discussion section: 

“In the Spanish context, previous research has also found effects benefits (on preventable causes of mortality) in areas of higher investment intensity where two or more urban regeneration programmes overlap (Zapata-Moya 2017 impact; Zapata-Moya 2021).” … “This result contrasts with the null effects found in Denmark for the years 2015 to 2020 (Nygaard 2023), and with the favourable evolution of residents in urban regeneration areas in Andalusia (Spain) during the years 2008 to 2015 (Zapata-Moya 2021).”

We did not consider the papers from Smith, 2012; and Cummings, 2012; because they focus children from 11-12 years old, while we study the adult population above 14 years old (the population target has been added in methods). The same with Rodger’s paper that investigates effects on injuries, cardiovascular and respiratory conditions. 

3. In the introduction, the authors refer to the lack of prospective studies as a factor that raises a number of weaknesses in studies on the health impact of urban regeneration programs, and mention specifically the following (Changes in the composition of the population; Failure to consider confounding factors and Previous trends). Although the methodological effort made by the researchers to develop a quasi-experimental approach is commendable, the cross-sectional nature of the data used does not allow in any case to overcome the three limitations mentioned above, especially the risk of attributing effects to the intervention without controlling for the population change that occurred before and after the intervention, as well as the study of previous trends in the most vulnerable neighborhoods. Combining the quasi-experimental method with the individual-level propensity score technique is an alternative to try to reduce these risks; however, the specialized literature indicates that there is a greater population change in neighborhoods intervened by urban regeneration programs. The researchers try to show that these differences in population change are not operating in their analyses, but I believe they should acknowledge that having only two cross-sectional sampling points does not allow for a categorical assertion of the non-existence of effects due to the population change motivated by the interventions.

Thank you for recognising the methodological efforts to overcome cros-sectional limitations. 

We agree with the reviewer that results have to be taken with caution, as they cannot be simplistically attributed to the intervention. We have expanded the limitations suggested by the reviewer in this point 3 but also in point 4, with the following paragraph: “This study has several limitations. Caution should be exercised when interpreting the results causally. Even though we used comparison areas and propensity score to make the distributions of population features between comparison and intervention groups comparable, threads to causality could arise if other factors differently affect both groups in previous trends or during the intervention. If the health indicators between the intervention and control groups do not move in parallel in the previous periods, this would imply that factors other than the intervention are at work and the results obtained could not be attributed to the intervention. For example, if the lasting effects of the 2008 economic crisis had affected the intervention group more severely than the control group, this could have further worsened their health indicators in 2016. In such a case, these indicators would tend to return to normal in 2021 and the improvement would not be attributable to the intervention. Another circumstance that could occur is that the intervention itself has changed the social context, such as a process of gentrification, and therefore these improvements in health are observed and cannot be attributed to the intervention. However, we believe that these effects, even if they exist, should be limited and do not compromise the overall results.” 

4. In addition to the limitations mentioned, there are external factors (the 2008 economic crisis and subsequent recovery, which some studies place from 2016 onward) that could be affecting the results, especially in those more vulnerable neighborhoods, as there could be a “regression to the mean effect" after a greater worsening of the main outcomes in the crisis phase. This could be happening especially among women, as the authors indicate; these showed worse health outcomes than men before the intervention, and therefore, there would be more possibility of a tendency (beyond the intervention) to improve, a "regression to the mean effect" among a particularly vulnerable group in a context of vulnerability. To rule out this alternative explanation for the main findings of the study, trends in these indicators should be explored jointly in intervened and control neighborhoods whenever possible.

We have grouped this topic with point 3 in order to highlight and expand on these limitations.

5. It is also important to mention another external factor that could influence the results in some way: since the COVID-19 crisis and lockdown occurred in 2020, the researchers should at least comment on how this could be impacting the potential effects found on mental health, self-perception of health among women, and alcohol consumption. Similarly, with the increase in the use of psychotropic drugs among men.

A paragraph on the potential lasting effects of the economic crisis and COVID-19 has been added with the with the recognition of conditionality of results:

“We must bear in mind that during the years under study, there have also been two relevant circumstances. On the one hand, the long-lasting effects of the economic crisis during 2016 and the emergence of the COVID-19 pandemic in early 2020. Both events have led to a worsening of mental-related health problems, especially among the male population during the economic crisis and among young adult women during the lockdown of COVID-19. It is difficult to identify to what extent both crises may have differentially affected our comparison and intervention group and through which mechanisms. If this were the case, it would be a limitation of our research. There is evidence of oscillations in health inequalities during both crises but in the medium term they translate into persistence of inequalities (Bartoll-Roca et al 2017; Martinez-Beneito et al., 2023). For this reason, we believe that the effects of both crises on the different evolution of health status between the intervention and control groups are of small amount.”

6. Regarding these external factors, the researchers in the discussion section point out as a strength of the study that "Indeed, any other contextual factor affecting the results at the city level should be equally affecting all the groups studied." In my opinion, this is a risky statement, as there is no guarantee that the impact of the economic crisis, as well as its recovery period, and even the COVID-19 crisis (as external factors) will have a similar effect among all city groups and all neighborhoods. Also, the potential impacts of the programs on health could be conditioned by the differential effects of the economic crisis.

Zapata Moya, Á. R., & Navarro Yáñez, C. J. (2021). Urban regeneration policies and mental health in a context of economic crisis in Andalusia (Spain). Journal of Housing and the Built Environment, 36, 393-405.

Certainly, this condition cannot be expected meet for sure and we have supressed the sentence. Regarding the potential effects of COVID-19, we refer to the added paragraph in point 5.

7. In a more nuanced interpretation, I think it is essential to recognize, especially in the discussion of the article, the potential for conditional effects of the intervention depending on the contextual characteristics of the neighborhoods.

We have added in the discussion the statement: “In any case, the effects of the intervention must be interpreted conditional on the contextual characteristics of the neighborhoods.”

8. Regarding mental health, the researchers choose to dichotomize the scale variable from a score of 3, considering that from this score onwards, there is a risk to mental health. I believe that this necessitates a more detailed explanation and references from reputable sources to support the authors' decision.

Different counting for Likert scales is possible for the GHQ-12. However, the coding (0, 0, 1,1) and dichotomization at the sum of 3 scores is widely used and a recommendation in the literature. Here you can find some examples: 

Piccinelli M, Bisoffi G, Bon MG, Cunico L, Tansella M. Validity and test-retest reliability of the Italian version of the 12-item General Health Questionnaire in general practice: a comparison between three scoring methods. Compr Psychiatry. 1993 May-Jun;34(3):198-205

9. The explanation for why only positive effects are observed among women refers to a conventional argument that rests on the idea that women have greater exposure to community life in the neighborhood and, therefore, to the interventions of these projects. Perhaps the discussion could be enriched if the authors delve deeper into this matter and are able to reference various explanatory pathways (which may be complementary) from the perspective of the sociology of health.

Thank you for the suggestion. We see the point of the reviewer, but we would prefer to keep the current explanation in order to not exceed the limit of words of the manuscript.

Regarding the style of presenting information and the article's wording, I find it suitable for publication after addressing the points and suggestions made earlier.

6. PLOS authors have the option to publish the peer review history of their article (what does this mean?). If published, this will include your full peer review and any attached files.

Do you want your identity to be public for this peer review? For information about this choice, including consent withdrawal, please see our Privacy Policy.

Reviewer #1: No

Reviewer #2: No

Reviewer #3: No

---

## [Decision Letter · Decision Letter 1]

28 Feb 2024

Short-term health effects of an urban regeneration programme in deprived neighbourhoods of Barcelona

PONE-D-23-31700R1

Dear Dr. Bartoll,

We’re pleased to inform you that your manuscript has been judged scientifically suitable for publication and will be formally accepted for publication once it meets all outstanding technical requirements.

Kind regards,

Angela Mendes Freitas

Academic Editor

PLOS ONE

Reviewers' comments:

Reviewer's Responses to Questions

**Comments to the Author**

1. If the authors have adequately addressed your comments raised in a previous round of review and you feel that this manuscript is now acceptable for publication, you may indicate that here to bypass the “Comments to the Author” section, enter your conflict of interest statement in the “Confidential to Editor” section, and submit your "Accept" recommendation.

Reviewer #1: All comments have been addressed

Reviewer #2: All comments have been addressed

Reviewer #3: All comments have been addressed

2. Is the manuscript technically sound, and do the data support the conclusions?

Reviewer #1: Yes

Reviewer #2: Yes

Reviewer #3: Yes

3. Has the statistical analysis been performed appropriately and rigorously? 

Reviewer #1: Yes

Reviewer #2: Yes

Reviewer #3: Yes

4. Have the authors made all data underlying the findings in their manuscript fully available?

Reviewer #1: Yes

Reviewer #2: No

Reviewer #3: No

5. Is the manuscript presented in an intelligible fashion and written in standard English?

Reviewer #1: Yes

Reviewer #2: Yes

Reviewer #3: Yes

6. Review Comments to the Author

Reviewer #1: A general map of the city shall be adecuate to undestand some of the results shown. It is an interesting approach to a trendic topic nowadays.

A Planning Handbook has been recently publicated by Spanish Ministry of Health: "Guia para planificar ciudades saludables" 2020, (Fariña et al.,) may be it will be interesting to mention it in the introduction.

Reviewer #2: (No Response)

Reviewer #3: The authors have addressed most of the comments I have made, with the exception of the "explanatory" interpretations of why the results may be among women and not men. This is an issue of utmost interest that affects the differential benefits of "flexible resources" under root cause theory. I understand that the authors prefer not to go beyond words, but this is a scholarly effort worth considering, since it is an explanatory, rather than a descriptive challenge of potential differential impacts by gender.

7. PLOS authors have the option to publish the peer review history of their article (what does this mean?). If published, this will include your full peer review and any attached files.

Reviewer #1: No

Reviewer #2: No

Reviewer #3: No

---

## [Editor Report · Acceptance letter]

5 Apr 2024

PONE-D-23-31700R1 

PLOS ONE

Dear Dr. Bartoll, 

I'm pleased to inform you that your manuscript has been deemed suitable for publication in PLOS ONE. Congratulations! Your manuscript is now being handed over to our production team.

Kind regards, 

on behalf of

Dr. Angela Mendes Freitas 

Academic Editor

PLOS ONE